# Cultural Bereavement and Mental Distress: Examination of the Cultural Bereavement Framework through the Case of Ethiopian Refugees Living in South Korea

**DOI:** 10.3390/healthcare10020201

**Published:** 2022-01-20

**Authors:** Myeong Sook Yoon, Nan Zhang, Israel Fisseha Feyissa

**Affiliations:** 1Department of Social Welfare, Jeonbuk National University, Jeonju 54896, Korea; yoon64@jbnu.ac.kr (M.S.Y.); sishou1122@hotmail.com (N.Z.); 2School of Global Studies, Kyungsung University, Busan 48434, Korea

**Keywords:** cultural bereavement, cultural bereavement interview, mental distress, refugees, culturally distant host countries

## Abstract

In South Korea, a mono-ethnic nation, refugees and asylum seekers from culturally distant countries are exposed to cultural bereavement, cultural identity shock, and cultural inconsistency for themselves and their children. Along with biological, psychological, and social factors, this phenomenon is hypothesized as playing a major role in an increased rate of distress among refugees. This study explored the experiences of 11 Ethiopian refugees living in South Korea, and their relevance to cultural bereavement while affirming and suggesting an update for the cultural bereavement framework. The analysis showed the refugees experiencing a slight continuation of dwelling in the past; a sense of guilt due to the fading of one’s culture; different types of anger; and anxiety with relation to the cultural identity of themselves and of their young children. Strong religious beliefs, a continuation of religious practice, informal gatherings within the Ethiopian diaspora, and organized community activities provided an antidote for cultural bereavement. The implication of the result hopes to assist and direct practitioners to identify complex manifestations of mental distress that often get wrongfully labeled as to their causation as well as methods and sources of diagnosis. Any update on the cultural bereavement framework also needs to consult the setting and peculiar circumstances of the displaced people in question.

## 1. Introduction

In the process of migration, the migrant is subjected to stress. However, for a refugee who is involuntarily migrating, the stress will escalate much faster as there will be less or no preparation to migrate to the new location. Normally in the first stage of migration, a stage of deciding and preparing to move, any migrant might face relatively lower mental distress. Whereas, in the second stage (physical relocation) and the third stage of post-migration, mental stress will build up and reach a critical point [1,2,3]. Especially, the third stage of migration will have its toll as the migrant will be expected to learn new socio-cultural rules and assume a new role [1,2,3]. The level of stress on forcefully displaced persons could thus be hypothesized to be much higher. 

Studies are exposing the effects of traumatic migration experience on displaced populations such as asylum seekers and refugees who are probably fleeing from torture, war-related trauma, violence, or persecution [4,5,6,7,8,9]. Deducting from these studies, the migration experience of the refugee and asylum seekers is highly associated with mental health problems (PTSD and depression) at its core. Lower mental health was also observed in people who migrated to a different country than those who remained in their own country [5,7,8].

Within the study of migration and mental health, Bhugra and Ayonrinde argue that the personality structure of an individual who is forcefully displaced and persecuted could be highly exposed to mental distress during the pre-migration stage [10]. Here the assumption is that the personality structure of the refugee is embedded in cultural identity and the refugee experience will tamper with personally held cultural norms and result in cultural incongruity [1]. In other words, as personality is directly affected by cultural factors [10], culturally shaped personality influences the person’s patterns of interaction in the new society [11,12]. Cultural Bereavement, culture shock, a discrepancy between expectations and achievements could be factors for distress in the migration stage [1,13,14]. In the post-migration stage, acceptance by the new host nation or culture could be a potential factor [1]. 

As an alternative cultural perspective on the refugee experience, Maurice Eisenbruch’s [15] cultural bereavement approach provides a better fitting framework for understanding refugees’ distress. Cultural bereavement is a type of grief reaction primarily caused by the uprooting and loss of one’s home, cultural values, social networks, and identity [1,15]. In Eisenbruch’s [15] concept of cultural bereavement cultural loss is understood as ‘the person’s or group’s experience of losing familiar social structures, cultural values, and self-identity’. According to his study, cultural bereavement could be a “display of feelings which is characteristically identified as a continued dwelling in the past; being visited by supernatural forces from the past while asleep or awake; suffering a feeling of guilt over abandoning culture and homeland; feeling pain if memories of the past begin to fade but find constant images of the past (including traumatic images) intruding into daily life; yearns to complete obligations of the dead; and feeling of stricken by anxieties, morbid thoughts, and anger that mar the ability to get on with daily life.”

Prior studies solidified the relationship between mental distress with the plight due to cultural bereavement. However, the interpretation of this relationship is complicated due to ‘the misinterpretation of the cultural expressions of grief by Western-trained clinicians and the Western diagnostic criteria of psychiatric disorders that may not be applicable in people of different cultural backgrounds [1]. Additional studies have then discussed the need to integrate a culturally sensitive understanding of social, cultural, and economic factors in the treatment of displaced people’s mental health issues, rather than just sticking to Western diagnostic criteria [16,17,18]. Effective diagnosis in such cases is advised to underscore elements like religious beliefs, cultural expressions of grief, and cultural rituals while analyzing issues surrounding displaced persons [19]. Eisenbruch in his cultural bereavement study with Southeast Asian refugees, stressed the exploration of religious belief and practices, stressing the importance of “traditional” treatments in the bereaved immigrant population [15]. Again, Eisenbruch’s CB interview also explores the language and cultural elements of the bereaved person [20].

Cultural understanding of refugee stress is thus directed towards exploring experiences and expressions of distress which are all embodied within the cultural identity of the displaced person. Cultural identifiers like religion, customs, language, dietary habits, etc. of the refugee may not necessarily be identical to the host country’s cultural expressions and could potentially be the reason for distress during cultural contact. Specifically, hazy concepts of social status and discrepancy between aspiration and achievement are mostly defined culturally and could result in poor self-esteem, leading to depression [11]. Above all, it should also be remembered that social and cultural values and behaviors are generally more resistant to change and are usually the last element to be adjusted during acculturation [21].

The experience and distress of the displaced person will thus be better understood by the incorporation of cultural bereavement with the mediating effect of cultural identity. Understanding the cultural elements of the bereavement process will also create a wider room for diagnosis and much better psychological treatment. According to Einsenbruch [15], incorporating cultural bereavement perspective can reduce the probability of refugees mistakenly diagnosed as having mental stress while their “symptoms” indicate deep social distress, the perception, meaning, and language of which are culturally defined. Most importantly, the disorder can be observed in refugees that do not have psychiatric signs in western terminology [1]. The incorporation may also enhance the diagnosis of co-morbid conditions, thereby enriching health care. It will also change the therapeutic emphasis from diagnosis to prevention by emphasizing the preservation of cultural meaning.

This qualitative study explored the cultural bereavement experience of a culturally distant refugee group living in South Korea—Ethiopian refugees. Within a mono-ethnic country like South Korea, refugees are potentially ex-posed to cultural bereavement, cultural identity shock, and cultural inconsistency which can instigate increased rates of distress. In this study, the relevance of the migration experience of the Ethiopian refugees to cultural bereavement is investigated.

### 1.1. The Ethiopian Refugees’ Context

South Korea is not a common destination for Ethiopians to seek refuge, but it appears a fact that more and more (although the number remains low) Ethiopians are seeking asylum in South Korea since 2010. For instance, in 2017 alone there were 62 asylum applications, of which 23 (46.9%) were accepted. In fact, the post-1990’s Ethiopian emigration patterns show the top destination countries were Sudan, the United States, Israel, Djibouti, Kenya, Saudi Arabia, Canada, Germany, Italy, and Sweden [22]. The study participants fled Ethiopia during the internal political instabilities in Ethiopia between the 1990s and 2010. They attributed different instances and circumstances as reasons for exile, but all their reasons pointed towards the political instability and persecution instigated by the government ruling at the time.

Within the study of migration experience being relevant to cultural bereavement, this particular group of refugees in South Korea is an interesting case for two reasons. First, South Korea is a 97% ethnically homogenous society, having a low number of immigrants and refugees living in the country. Ethnic density was mentioned as a factor that influences the rates of mental distress in ethnic minorities if ‘the size of a particular ethnic group is few in proportion to the total population’ [1]. Additionally, a sense of alienation may occur if the cultural and social characteristics of an individual differ from those of the surrounding population, whereas a ‘sense of belonging tends to occur if the individual and surrounding population have similar cultural and social characteristics [1]. The cultural distance, the low Ethiopian ethnic density, and their experience of resettlement will pose distinct psychological, cultural, and issues of mental distress. 

Not only the low density of Ethiopian refugees, but the intercultural relationships with the native Koreans also have an element of discrimination that might exacerbate their problems as refugees. For instance, although showing a great improvement from the previous wave of surveys, the 2017–2020 seventh wave of world value survey among 1245 Korean participants shows that 13% of them mentioned they don’t want people who speak a different language as a neighbor; 15% mentioned they don’t want people of a different race as a neighbor; 22% of them mentioned they don’t want immigrants as their neighbor [23].

Second, South Korea’s refugee acceptance rate is below the international standard. In addition, the structural adequacy in dealing with issues of asylum seekers and recognized refugees is in its infancy. Although there have been asylum applications since 1994, the country accepted its very first refugee in 2010; which could be an indication that the resettlement journey for any asylum seeker is faced with institutional inadequacies [24]. Thus, Ethiopians’ post-migration experience from the point of entering the country to the point of acquiring recognition as refugees could be hypothesized as having relevance to cultural bereavement.

### 1.2. The Objective of the Study

Although it has been commonly referred to in different works of literature, cultural bereavement in a refugee setting appears to have been accepted rather than explored further. The idea of cultural bereavement was suggested to be carefully tested in further clinical and ethnographic research. The earlier study of Eisenbruch stressed that “a range of ethnic and cultural groups needs to be compared, including groups who have experienced various stressors; refugees at each stage of the life-cycle; refugees in post-migration stages; those who have resettled in various countries with a range of resettlement policies; and those in refugee camps” [15]. Hand in hand with cultural bereavement study, cultural bereavement interview (framework) was originally developed to explore reactions to personal losses and to losses of both the social systems and the cultural meanings [20]. Since it was introduced, however, it was less tested in various refugee groups who are entering culturally distant host countries. It, therefore, appeared important for this research to investigate the relevance of cultural bereavement leading to mental distress within the migration experience of culturally distant Ethiopian refugees living in South Korea.

The primary research question is ‘have the experiences of displacement and living as an asylum seeker or refugee in South Korea resulted in cultural bereavement?’ This question aims to identify the relevance of the refugee experiences to cultural bereavement and to identify if there are coping mechanisms that contributed to positive well-being. Adding to that, the study aims to inform practitioners to incorporate cultural meaning, manifestation, and expression of mental distress when dealing with subtle traumas such as identity loss or cultural bereavement of a refugee. From an academic point of view, this particular investigation also aims to add value to the cultural bereavement framework in the refugee context and how it should be revised for utilization or adaptation.

## 2. Materials and Methods

### 2.1. Qualitative Design

A semi-structured interview guide was used for data collection. It was developed using the cultural bereavement interview [25]. The cultural bereavement interview explores reactions to personal losses and losses of both the social systems and the cultural meanings. Areas explored in the CBI are:Memories of a family in the homeland and clarity with which appearance of relations is recalledContinuing experiences of family and pastGhosts or spirits from the pastDreamsGuiltStructuring of the past in the homelandPersonal experiences of death and funeralsAnxieties, morbid thoughts, and anger in response to separation from the homelandThe comfort derived from religious beliefsThe comfort derived from participation in religious gatherings.

### 2.2. Recruitment Process

Potential participants were contacted through the administrative body of the Ethiopian community in Seoul, South Korea. Eligibility criteria included refugee men and women, aged over 18 years, and having lived in Korea for at least a year as a recognized refugee. Based on South Korean refugee status determination process, a minimum of 1 year as recognized refugee was considered a reasonable minimum amount of post-exile experience for the participants. Each potential participant was informed of the purpose and process of this study, their rights as participants, and the researchers’ obligations. Those who were interested in participating were instructed to contact the lead author via email, phone, or text message. The necessary Institutional Review Board approval was also maintained from Jeonbuk national university (IRB FILE No: 2020-08-001-002) before recruiting the participants and the actual data analysis.

### 2.3. Study Setting

Interviews were conducted in the participants’ homes twice, and each interview lasted 2–4 h. Participants voluntarily provided a written indication of informed consent, then completed a short survey examining their socio-demographic information. Interviews were conducted mostly in the Amharic language. All participants reported they were comfortable with either Amharic or English for the interview. The interviewer was an Ethiopian national who is among the research team for this study. He is fluent in Amharic and English.

The interviewer used an interview guide for the cultural bereavement interview (CBI) that contained 11 questions relating to migration experience and cultural bereavement. These questions were tested in a pre-interview with one refugee who was not included among the interview participants. Example questions are as follows: “What do you imagine/remember when thinking about your family back in your home country? Who and what is missed?”; “Many people can feel that members of their families they left behind (or those who died) are still with them, sometimes they can hear, see, feel, touch or even smell them, or feel a ‘sense of presence’ as if they are around. Have you had any of these experiences?”; “After being away from home for a while, some people find it hard to remember what their homeland, family, or friends look like. Do you have any difficulties remembering what people or places look like?”; “Sometimes if refugees do not think about their homeland, friends or family as much as before they can feel guilty. Others can feel guilty that they got out, and started a new life. Has this happened to you?”; etc.

The interviewer engaged in a flexible, semi-structured interview process using probes to clarify information relating to key themes and allowing the participants to change the course of the conversation and bring up relevant issues. All interviews were digitally recorded with permission. The participants were also told that if they wish to do so, they could stop the interview at any time. The recordings of the interviews were transcribed verbatim in Amharic and literally translated into English text documents.

### 2.4. Data Analysis

Using Hyper research 4.0.1 software, initial data analysis involved immersion in the data as a whole: reviewing, categorizing themes, and analyzing the complete collection of transcripts. We then performed an analysis noting recurrent topics and themes and recurring similar issues. The method this study used is template analysis. Template analysis is described as a “group of techniques for thematically organizing and analyzing textual data” [25]. It is a well-established qualitative method for exploring people’s experiences [26,27].

Data were examined not as exact representations but rather to elicit the participant’s interpretations of their experiences and their social worlds. The focus of the narrative analysis was on the connections between their migration experience as a refugee, the experience of cultural bereavement, and any coping mechanism during the settlement process.

Drawing on the concepts of cultural bereavement and its associated migration experience and coping mechanisms, thematic narratives were identified through discussion and building consensus among the authors. Throughout the analysis, the authors engaged in discussions on the meanings, interpretation, and overall intellectual understanding of the issues at hand. The author’s epistemological perspectives, as well as scholarship standpoints, were utilized throughout the data analysis. All the participants are represented by a number throughout the data analysis.

### 2.5. Participants

Eleven refugees, seven men and four women were selected for the study (see Table 1). Their age ranged from 31 to 56. Their length of stay in South Korea ranged from 3 years to 20 years. However, their length of stay as a refugee ranged from 1 year to 12 years. Three participants also attained a permanent foreign resident status. Four participants were living with their family and the remaining seven were living alone. Eight participants are followers of orthodox Christianity and the remaining three identify as protestant Christians.

## 3. Results and Discussion

The eleven areas of the CB interview were explored to capture a recurring theme within the transcribed inter-view data. The findings supported the experience of cultural bereavement within the Ethiopian refugee population. With contextual modifications, however, the CB framework is also able to illuminate the challenges and the coping mechanisms of our study participants. Here the results of the investigation and a discussion are presented with key themes and participants’ quotes in Table 2.

### 3.1. Memories of a Family in the Homeland and Clarity with Which Appearance of Relations Is Recalled

In this section of the interview, the participants were asked what they imagine/remember when thinking about their family back in their home country: what, who, and when they missed, and about events that trigger their memories. Memories about family members, friends, family gatherings, family and friend social events, and neighborhoods were mentioned by participants when they think about family and home in general.

The memories about places, family neighborhoods, and names of places seem to fade as the duration of exile increases. This was highly evident for P6, P7, and P8. They attested they forgot the names and ambiances of some of the well-known places around their neighborhoods. When presented with some recent photos of some of the places they should know, they found it hard to identify them correctly. For these participants, their recollections about their parents and family members seemed intact only around memorable significant life events, failing to remember details of those events. Longer exile experiences led to fading or less vivid memory of what places (neighborhoods) looked like. This vanishing memory and a constant change in Ethiopia since they left is also becoming a threat to the participant’s sense of belonging.

Family members and family/social gatherings were mentioned among the things the participants missed. The weak social connectedness that the participants encountered in their exile made them miss casual family gatherings. Almost all participants mentioned missing the custom in Ethiopia for family members to eat food together from one big plate. The custom is believed to solidify family closeness and familial ties.

Ethiopian national holidays, weak social connectedness, Ethiopian television channels, and any news about Ethiopia were mentioned as triggers for memories to reappear. However, P6 and P7 mentioned having trouble relating to the changes that they watched within the television programs.

This section also looked into the clarity the participants have with regards to the appearance of relations. The assumption is that, after being away from home for a while, they will find it hard to remember what their homeland, family, or friends look like. The question revolved around if they have difficulties remembering and if they do anything to stop the memories from fading. In this particular question, a participant in exile for 20 years expressed an insightful view of the effects of long-term exile away from home. He mentioned:
“…I don’t know if my memories are in the right place… I feel like they are fading. Places I do remember 20 something years back still exist today. I can’t even remember some of their names… I do have a good memory of my parents and family members though. I feel like Childhood friends and memories are getting patchy. I lived almost half of my life here…what do you expect? I even have a Korean name now. Whenever I get to see [on TV] how things change Ethiopia I’m a foreigner to it…”

He also added:
“…I don’t see myself trying to retain those memories in any special way. I’m also making memories here. It’s a fact that the more I live here I’m becoming more like Korean than Ethiopian.”

The other participants mentioned that tuning to Ethiopian TV channels and video calls with a family member do help them to keep up with family and major events in the country. In a common tone, however, most of them mentioned these keeping-up mechanisms don’t feel as real. In other words, although they feel grateful for the means of contact, they still feel the distance. Virtual means of communication (TV, social media, etc.) with family and culture could be an instant antidote for cultural bereavement. However, based on participants’ experience, capturing what they consider theirs through virtual means could exacerbate their feeling of loss.

### 3.2. Continuing Experiences of Family and Past

The interview in this section was intended to understand if the participants had an experience of a feeling that members of their families they left behind (or those who died) are still with them, or if sometimes they can hear, see, feel, touch, or even smell them, or feel a ‘sense of presence’ as if they are around. For some reason, this question didn’t seem to sit well with most of the participants, with most admitting that they had no experience similar to this, except minor things reminding them of their families and friends. However, it was mentioned that these reminders appear always unpleasant for immediate families who are separated.

Above all, the realization that they might not go back to their country and families marred some of the participants’ feelings to remember their past. The disruption to interpersonal bonds as a result of their exile created a reaction of grief, nostalgia, and homesickness. The participants attested that these feelings might compromise their current life. Refugees who are separated from their immediate family usually use any available means to reunite with their family in South Korea in the possible shortest time. Most of them did invite their immediate family members as soon as they got their refugee status. Participants’ grief towards loss of social bonds, status, or cultural identity was managed by continuous acceptance of their actual circumstances and through the determination to create a better life for themselves and their families.

### 3.3. Ghosts or Spirits from the Past

This section asked the participants to share if they have experience of ghosts or spirits of the dead family member or someone close visiting them. Like the previous question, this question was also unpopular. The concept of ghosts and spirits is taboo for Ethiopians. Eisenbruch [15] suggested ‘the uprooted person… continues to live in the past, is visited by supernatural forces from the past while asleep or awake’.

However, in Ethiopian refugees’ case, it is difficult to prove word by word that their experiences fully relate to this presupposition. In other cross-cultural study settings, it is therefore important to reconsider the wording ‘ghosts or spirits from the past’ into something culturally appropriate. The focus on ghosts and spirits from the past, for example, is not a cultural universal. This is also problematized in the literature on CB, and ghosts should be defined in a way that has more of a transversal cultural logic to it. Especially the Orthodox Christian participants consider ghosts or spirits of this kind as something to be avoided or not to think about. Orthodox Christians of Ethiopia, as some of the refugees here do, hang a picture of Christian saints as an amulet to protect themselves from ghosts or ‘bad spirits’ as they call them. With this particular question, it is safe to say that, if there had been such an experience of encountering ghosts or spirits, the participants are not comfortable enough to discuss it freely.

### 3.4. Dreams

The participants all confirm at least having one or two vivid dreams a month which in any way are related to their country, family, or something that they left behind. This section also asked if there are triggers to dreaming; like worries or nostalgia. One of the triggers for worries and nostalgic experiences was attributed to weaker social ties the participants have with local Korean people. The more participants felt like outsiders, the more they began to yearn for stronger social connections they have with a family and friends. (P7) responded:
“…You will be nostalgic here… The system here doesn’t let you create a new identity. You can ask all of us here. How many of us have a real Korean friend? How many of us know a Korean that we invite to our house or get invited to theirs to socialize? No one. Nobody established a meaningful relationship with Koreans that is bigger than being a work colleague. If it were in another country wouldn’t that be a different story? I don’t know. What I hear from friends in America and Canada is all about having more stronger ties with each other and creating a meaningful relationship with local people. If at all, life in Korea will make you more nostalgic… always wondering what situations would have been if you are in familiar and friendly places…At least dreaming about the good old days brings joy [giggle]…”

Some urgencies of the participants also appeared in some of their dreams. In Ethiopia, there is a customary obligation to visit a bereaved family even long after the funeral; if the person is a relative or close friend and was unable to visit within the three days after the burial ceremony. P11 who wants to comfort her friend’s family for their loss of a child mentioned what her dream experience was like:
“… In my dream, I always think that I’m back home and in the middle of it, there is always an urge to visit my best friends’ family. They recently lost their older son and I couldn’t attend the funeral because I was here… So within my dreams, I always dream of a moment that I went back home and urged myself to go and see this family…After that, I wake up and know it’s not going to happen…”

Dreams are mentioned between two extremes of bringing desperation and sources of comfort. For participants separated from immediate family members, the degree of worry and experiences of dreams are painful. For others, these dreams are sources of sweet nostalgic experiences about their past and identity. It is also mentioned that there is sometimes a slight sense of disorientation as soon as they wake up from such dreams and realize that they are in a different country.

### 3.5. Guilt

This section of the interview asked the participants if they feel guilty if their memories are fading about their homeland, family, or friends. The interview in this section intends to understand if there is a guilty feeling that is associated with these refugees fleeing their homeland and starting a new life. Again, within this section, the participants were asked why they left their country.

There is an underlying guilty feeling in general. In a situation where they are reminded of their exile and how their situations turned out, they mention they would have preferred not to leave in the first place if it hadn’t been a threat to their security. Guilty feelings were also associated with the loss of social status. P(5) mentioned:
“…I regret leaving my country and my achievements behind. I was working as a journalist and it was because of a politically critical book I wrote that the Ethiopian government targeted me. I saw my colleagues paying the ultimate sacrifice for what they believed in… Getting arrested, fighting the political system in any way they can… I knew they [the Ethiopian government] are coming for me and I guess I was not as strong as my colleagues. I thought about what will happen to my wife and two kids if I get arrested. I had to take any opportunity to leave the country and make sure my family was safe. We are safe now… that’s what matters, I guess.”

In the same line of thought, P(4) added:
“…Not only for me, but I think for all refugees here, there is always a 60% opportunity and a 40% risk leaving one’s country. That’s true for me too. I thought about having safety and protection as an opportunity and uncertainties of starting a new life as a risk. You don’t have time for guilty feelings when you are on the run… But after you settled somewhere and know you are safe and protected, the guilt starts kicking in. That’s the risk you have to swallow… You probably might not go back and see your family, country, or whatever.”

Guilt appears a common feeling all of them felt and expressed the idea that they knew this feeling would be part of their new life. Feelings of guilt are unavoidable, but something they would rather not think about, though with some of the participants, there is a sense of hope that they can reconnect with people they have left behind. P(11) mentioned:
“…I always regret the nephews and nieces that I never get to meet…but who knows one day we might meet someplace… These are some of the things which you will leave for ‘hope’.”

### 3.6. Structuring of the Past in the Homeland

This section of the interview intended to understand how Ethiopian refugees remember their past; their memories and if there is any sentimental thing that they brought with them.

As political refugees, the common sentiment among the participants is the fact that their memories are shaped by the political system that pushed them out of the country. Almost all of them mentioned a broken political system that made their lives harder from time to time, which in return made their recent memories highly unpleasant. P(2) articulated: 

“…If it was not for my family, everything I remember is how the social system was broken and how it makes us [the youth] hopeless. Until now, I feel like the country wronged me… To some extent, I feel like I’m lucky to be here. Although, I can’t deny that I miss my family and friends… Once or twice a day you will browse through Facebook and see their faces… major events of family and friends are also posted on Facebook and you somehow feel like part of just by watching and commenting on their post.”

The tendency to collect sentimental things before exile seemed to differ from participant to participant. Some of them had the chance to bring something, others didn’t have the chance, and the rest didn’t even think about bringing sentimental things.

P(5) carefully collected what he thought was important before his exile. He mentioned his memories and his attachment with sentimental things he brought with him like this:
“I often think about the life I lived back home, especially when life gets tough here. My memories are happier and more stable… I had everything I needed. I think my family was much happier. Of course, here, with a minimum wage job, you can make more money than what you would make back home. But you can’t buy everything with money. I carefully brought almost everything I thought would be sentimental [pictures, a book I wrote and other books, family home videos]. It gives me comfort when I see them…It tells me and my children that I was indeed somebody, someone important… they [sentimental things] especially help me when the work environment [the menial job he is working] belittles you.”

On the other hand, P(7) mentioned:
“I never had the chance to collect things… I knew I will be separated from my family physically but I never had the idea that we will be separated this long… if I knew I would have a second thought… I can say I don’t have anything that brings memories…you just think [dream] about them… and more than anything… I think, my family here distracts me from thinking about my past.”

### 3.7. Personal Experiences of Death and Funerals

In this section, the assumption on the CB framework is that refugees may have lost relatives in their homeland while they are in exile. Among the participants, even though there was a threat against their lives, no one experienced death on their journey. All of them are politically driven refugees, and the danger almost all of them experienced was either getting arrested or being tortured. After exile though, as the time in exile increased, the participants experienced death of a family member or friends.

It’s a common sentiment among the participants that, hearing the news of someone passing, and not being able to be there physically for funerals and other ceremonies is a painful experience. The common expression of grief in most parts of Ethiopia is expressed openly, loudly crying and wailing, saying the deceased’s name, and beating their foreheads and chests. Physically injuring yourself to express grief is common and even expected in certain areas of Ethiopia, and crying shows that the deceased was loved. When participants mention they didn’t mourn properly is an indication that their circumstance didn’t allow them to express their emotions well. Something similar to this type of experience however was mentioned by (P10). She expressed her experience as:
“I lost my dear friend within three years that we separated. She was like a sister to me. We lived together and everything… I remember calling her a week before her death. She was suffering from a heart problem. Her death was sudden and unanticipated. a day before her death I was wondering why she is not calling me. On the next day, while I was in the middle of a short lunch break, I learned about her death through a post on Facebook. My initial feeling was disbelief and tried to call her and ask around… The truth kicked in and I didn’t know how to mourn her death. You know how our work in the factory is… I wanted to cry but I couldn’t… I remember hiding my tears… If I cried as much as I want to, how would I describe what happened? The difficult times where I’m alone and the dreams that followed the next week with no one to talk to about what happened. In my dreams, I was fully convinced that she is alive as if her death is not real. The sadness and grief will continue in the morning… With this type of disbelief, I struggled for some weeks. I still have dreams though, but not as frequent… Sometimes I wish I never heard the news… It’s better not to know I guess.”

It’s a tradition in Ethiopian Christian communities that, if a family member or a close relative dies, community members visit and comfort the grieving family during a three-day mourning period before the funeral. This also gives relatives time to arrive and pay their respects. Some people may choose to sit in silence rather than talk with the grieving family. Attending funerals and after-funeral gatherings as well as comforting the bereaved by staying in their house for days is considered the most important aspect of the Ethiopian mourning culture. When comparing the mourning culture with Korea, similarly, the long-lasting Korean tradition of sangjo (literally translated as mutual help) exists when consoling the bereaved. Recent sociological observation, however, indicates how mutual aspects of funerals are becoming a financial burden for those who are supposed to visit the mourning family—which could lead funeral ceremonies to become the affair of family members and close relatives [28].

### 3.8. Anxiety, Morbid Thoughts, and Anger in Response to Separation from the Homeland

It is worthy to mention also the experiences of the participants are completely distinct from one another. It would be trivial to label all their experiences as dangerous, traumatic, and surrounded by inhumane incidents. Although every exodus is backed up by danger, hopelessness, and a search for a better life and safety, not every participant goes through the same type or level of ordeal. Some even lose their lives before reaching their desired destinations, while others were not only opportunistic enough to reach their destinations within a short period but were able to thrive beyond their expectations.

Generally, from the 1980s onwards, mass exodus as a form of refugee is not a new thing for Ethiopian asylum seekers. The push factors out of the country, political instability being the only one for the participants, individual reasons’ intricate dynamics are much far wider than political push. Whenever the opportunity presents itself, all of the participants used that chance to flee for safety, better life, and security.

The journey by itself caused feelings of stress and uncertainty with lots of expectation crises. Again, several differences between South Korea and Ethiopia were a cause of distress. First, South Korea was not a common destination for Ethiopian Asylum seekers before the year 2000, and that created a sense of lonesomeness in some of the participants who fled to South Korea before the year 2000. Secondly, for all the participants, differences in religion, sense of community, and cultural practices in South Korea appeared to cause difficulties in terms of how the participants navigated life in South Korea.

Anger was found to be prevalent amongst some of the participants. This anger resulted from several different factors: expectation crises whilst living in South Korea, the separation from family, and the ongoing political situation in Ethiopia. Especially anger towards the Ethiopian government which allowed this to happen to them was vented through their social media outlets. They are constantly angered because they think the problem that forced them out of their country still exists to this day.

For some of the participants, there wasn’t enough time to understand what was going to happen before fleeing Ethiopia. For the rest, the expectation is a vivid understanding that the host country can protect them and they can be better off. However, after a while, a sense of anger started to emerge for reasons other than leaving Ethiopia but for situations encountered within South Korea. (P2) mentioned: 

“How could you have expectations of a place where you never been to? At the time I just don’t know where but I have to flee… Fleeing was the first thing in my mind. Eventually, I felt angry about why my situations turned out this way. It was my wish to be with everyone and everything I knew but what can I do about it. I feel angry the most when I realize that I don’t belong here and when I realize that I’m in a foreign land and whatever I do I’m considered a second-class citizen.”

Anger because of threats to identity and role occurred as a result of previous roles, status, and qualifications being unrecognized in South Korea. Participants reported that they sought lower-skilled and lower-paid employment positions, as previous employment and qualifications were not recognized in South Korea. However, most of the anger is also associated with the gradual fading of the cultural identity itself. Their concern is highly related to the gradual fading of their identity or belongingness. Especially the fact that their own or their children’s self-conception, self-perception, nationality, ethnicity, religion, and social class are being redefined according to South Korean social settings. (P7) responded: “…My children, being half Korean, feel they are different from the other kids in the school and often feel an outsider or different. But I know they feel more Korean than Ethiopian though.” With this type of anger, participants admit to being overly attentive to what’s happening culturally in Ethiopia and trying to spoon-feed their children about Ethiopia.

There is also a continued and exacerbated anger for participants who left Ethiopia because of their political opinions. Their anger is highly associated with what’s happening politically in Ethiopia. Even though they admit they are now safe from any threats, they did not stop opposing or “struggling” against the system that pushed them out of their country. (P1) puts it:
“Most of us turned into full-blown government opponents. You can see what most of us post on Facebook. Most of it is either exposing what the government did to us or engaging in conversations that could discredit the government that wronged us.”

On the other hand, the anger was also translated into a determination to succeed in life. (P7) mentioned:
“The only way to keep your sanity after a while is by telling yourself that ‘home is where you are’ and focusing on the future and wellbeing of your family. It will keep your mind busy and you will realize that your country is your family. The more I focus on the future I tend to feel less angry… What matters now is I’m safe with my family and the future might also take us back home [Ethiopia].”

### 3.9. The Comfort Derived from Religious Beliefs

Religion is very important for all the participants. Eight of the participants are Orthodox Christians and the remaining three identify as protestant Christians. For the Orthodox Christians, the existence of the Orthodox Christian church in Seoul is something they are grateful for. The church allows them to keep up with their distinct rituals and create a space for their religious festivities.

The question of whether religion helpful got a resounding “yes” from the Orthodox participants. However, their answers were related to how their religious beliefs helped them personally. However, when it came to feeling less upset about losing their family and homeland, they mentioned religious rituals strongly bringing memories about family and homeland. (P2) mentioned:
“The grace of saint Mary and her son is the reason why I’m standing today. If it wasn’t for the protection of Saint Gabriel (Angel), I wouldn’t be alive today. My daily prayers are everything for me…”

(P3) also added:
“Keeping up with our Orthodox Tewahedo belief system is what makes us Ethiopian. Whatever the situations are for us here, we will continue to keep up with the yearly fasting while remembering days designated for saints and the dietary restrictions.”

For the protestant Christian participants, the wider availability of protestant Christian churches created a relaxed option for them to exercise their religious beliefs. They were also allowed to hold a service in their distinct language in some churches. (P8) mentioned:
“It always comforts me to know that my life is in the hands of God. I passed through those times [difficult times] by God and believing what the word of God is saying.”

Their cultural identity is highly associated with their religious beliefs and practices. However, concluding that their belief provided solace for missing their homeland and family is an understatement. The distinctive religious practices have an exacerbating effect. In parallel, although without the exact cultural ambiance, all of them are grateful that they can practice their religious rituals in a similar way to how they would have back home.

### 3.10. The Comfort Derived from Participation in Religious Gatherings

For the participants, supportive social gatherings are not necessarily religious gatherings. They mentioned a religious gathering that helped them to thrive, but they also mentioned non-religious gatherings that provided comfort and solace. One is a gathering and event that is organized by the Ethiopian community. The Ethiopian community has its own administrative body that provides a platform for Ethiopian residents (not only refugees and asylum seekers but the expats as well) to help one another. Even though it is less frequent, the community organizes events and special social gatherings. The participants mentioned these events being very meaningful and memorable. (P4) explained:
“The Ethiopian community is where we [refugees] get our help from. Whether it is language issues or any information we need, it’s through people in the community that we get most of our help.”

Informal gatherings are also mentioned as having the merits of easing their life hardships. A small Ethiopian restaurant in Seoul created an opportunity for informal gatherings to happen. The participant mentioned this place as a place where they momentarily feel they are back home. (P4) mentioned:
“Weekdays are busy but most of us are at least free on Sundays. Every Sunday afternoon we run to Zion [the Ethiopian restaurant]. It’s a place where we feel at home, eat authentic Ethiopian food, and have coffee the Ethiopian way…You could at least chat with your fellow Ethiopians with your native language, see an Ethiopian face, and hear Ethiopian music.”

However, churches are, for the Orthodox Christian participants and protestant Christian participants, a place where regular (at least weekly) meetings happen. These gatherings are mentioned as the only providers of meaningful social interactions. It is in these religious and non-religious gatherings that the participants feel kinship, support, and a sense of community.

## 4. Conclusions

As the idea of cultural bereavement was recommended to be carefully tested in further clinical and ethnographic research, this particular study set out to investigate the relevance of cultural bereavement leading to mental distress in a different refugee group—Ethiopian refugees living in South Korea. The results of the investigation provide evidence of how cultural bereavement explains Ethiopian refugees’ distress while living in South Korea and their coping mechanisms.

The very first studies regarding cultural bereavement that looked into the case of Cambodian refugees identified traumatic loss of society and culture and how the Cambodian refugees were obliged to adapt rapidly to a new country. There was also a suggestion that the case of Cambodian refugees could represent other displaced people [15]. Within the scope of our study, the case of Ethiopian refugees could partially resemble the case of Cambodian refugees. For instance, the level of trauma incurred during the loss of society is incomparable while the loss of culture and the distress that occurred during adaptation have a similar pattern. Our investigation within the case of Ethiopian refugees indicated the Ethiopian refugees experienced a slight continuation of dwelling in the past, a sense of guilt for the fading of culture, different types of anger, and anxiety in relation to their cultural identity and that of their young children. This study also affirmed the claim of both Eisenbruch [15] and Schreiber [19] who considered cultural bereavement as an expression of deep suffering that is impossible to bring back to existing psychiatric diagnosis or therapy. Within the Ethiopian refugees’ group, the expression of distress due to cultural loss might be misunderstood or misdiagnosed. The effectiveness of ritual healing or dependence on one’s religious belief is then fully confirmed as the case analyzed by Schreiber.

As an antidote to cultural bereavement, strong and sustained religious beliefs, continuity of religious practice, informal gatherings, and organized community activities were identified in the Ethiopian refugees’ case. Although these ritualistic religious beliefs and gatherings seem to be mentioned in earlier studies as relief for cultural bereavement, this study also recommends the inclusion of non-religious informal gatherings and the size of the diaspora in the mix. In as much as religious gatherings provide a solution, informal gatherings outside of religious meeting places also serve as an antidote for the culturally bereaved. For the Ethiopians, strong religious beliefs proved to provide hope and a sense of protection in the most vulnerable situations. The freedom to exercise their religious beliefs also helped them to retain their cultural identities while minimizing the trauma caused by the fading or loss of a culture. An organized community not only provides support in day-to-day life but also creates opportunities for cultural preservation and maintaining kinship.

Within the interpretation of cultural bereavement and the application of the cultural bereavement framework, stubbornly remembering, having frequent nostalgic experiences should also be perceived as an act of resistance of “counter-memory”, not just a symptom or the cultural expression of suffering related what they have lost. In this line of argument, there is a need to further question or interpret the interrelated issues of (cultural) bereavement, nostalgia, and refugees’ condition in light of resistance to bereavement. In addition, the role of virtual connectedness that social media are providing needs further investigation for their role of maintaining connection despite the difference in location.

The implication of the study is to affirm and update the cultural bereavement framework in understanding mental distress which is only culturally determined. Following the few previous studies, we hope that the results of this particular study will assist and direct practitioners to identify complex manifestations of mental distress in refugees and asylum seekers that often get wrongly labeled as to their causation and methods of diagnosis. Any update on the cultural bereavement framework also needs to account for the setting and peculiar circumstances of the displaced people in question.

## 5. Limitations of the Study

One limitation in this study is that the investigators were flexible with the participants so that they could give as many answers as they wished during the interview. Such interaction might create bias and could affect the credibility of our results. In order to avoid such bias, we suggest that similar future studies ought to employ prolonged observations and conversations with participants, but also to check whether the findings obtained are recognized by the informants as representative of what they think and feel.

## Figures and Tables

**Table 1 healthcare-10-00201-t001:** Description of participants.

Participant Number	Age	Gender	Length of Time in South Korea (yr.)	Live with or without a Family	Language Spoken	Educational Status	Religion	Refugee Status	Time since Refugee Status Attained (yr.)
1	34	male	9	Without	Amharic/English	College graduate	OC	F-2	2
2	32		6				OC	F-2	2
3	56		6				OC	F-2	2
4	32		5				PC	F-2	3
5	35		4	With	Amharic/English/ Korean		OC	F-2	1
6	50		20				PC	F-5	12
7	40		15				OC	F-5	10
8	45	female	14	Without	Amharic/English	High school graduate	PC	F-5	11
9	31		5				OC	F-2	2
10	33		6			College graduate	OC	F-2	4
11	32		3	With			OC	F-2	1

PC: protestant Christian; OC: orthodox Christian; F-2: long-term resident visa; F-5: permanent foreign resident visa.

**Table 2 healthcare-10-00201-t002:** Key themes arising out of the data and participants’ quotes.

Theme	Participant Number	Example/Quotes
Memories of family in the homeland and clarity with which appearance of relations is recalled	P6	“I don’t know if I have a clear memory of what my neighborhood is like. It’s been a while… my memory is vivid when it comes to how places look like. I sometimes forget the name of some places too. You can’t forget what your parents look like though. I imagine a lot has changed.”
	P5	“… Memories about places in Ethiopia seems to stick. As a journalist in the country, I had a chance to travel different places in the country… I don’t know if I’m ever to forget that experience.” “…Here you will be drinking beer alone… beer is not a social thing anymore for me. You would be traveling alone too. It would be a different story if it had been in my country… What can you do about it Huh? you just try not to let it get to your head.” “Every challenge here brings a trigger. But, I try to avoid becoming a person who dwells in the memories of the past…If you attach everything to your past you will lose what you have here and I don’t want that… Whenever those memories overwhelm me I tend to focus on what’s here and now and do better to have a better life experience for me and my family.”
	P9	“ Saturdays, in my family were almost revered as if they are religious holidays [giggles]…I have a big family and we always meet in our family house every Saturday… We all eat from the same plate…that’s why I’m always trying to recreate that here by inviting Ethiopians to my house on weekends. It doesn’t feel the same sometimes, though it helps. A Saturday without some type of festivity is unbearable but you will learn to go with the flow…. That is being, working on Saturdays and not even remembering it’s Saturday.” “…I watch Ethiopian Tv channels. It keeps me connected… Thanks to the internet, there is no major news that I’m missing… I think that helps me to keep up with what’s going on… I also call my family and see their faces. We exchange pictures and that helps as well. But sometimes calling my family or watching some of the pictures won’t help you that much… Everyone [me and other refugees] will have a desperate need to experience things in person… not through TV or video call. Sometimes, I almost cry after the call ended… Especially, if that day is a holiday or someone’s birthday.”
Continuing experiences of family and past	P7	“…There is a time that you will convince yourself that you might never see your family or friends anymore. It’s a hard thing to do but if you want to keep your sanity, that’s what you will do. We [Ethiopians] have our challenges daily… Thanks to them we are not bothered by those feelings. But I’m not saying I don’t dream about my family and friends. But my interaction with them is only inside the dream. I don’t let it interrupt me while I’m awake.”
Ghosts or spirits from the past	P9	“…I think that’s why we pray every day so that we don’t encounter bad spirits [ghosts]… I don’t know if any bad spirit could enter this house…”
Dreams	P5	“… In the first 5 or 7 months, all I was dreaming about was home and about my wife and children. Not once did I ever imagine my life would be in exile. Before coming here, all my ambitions and aspirations were highly linked with my country. Unfortunately, I became a refugee…In those days, I think it took a while for my head to admit that I am in a different country and I might never go back. Having my family and my two children here this year, at least, eased my tensions… Now, I’m beginning to define home or country as any place where you have your family with you.”
	P1	“…For a few years, I lived and worked in a factory where social life was almost nonexistent. It was all about work and work alone… In those times these types of vivid dreams were comforting. In those dreams, there is a sense of wholeness… Having people you love around you and not living your life as some type of machine… But one thing you should know this good feeling is happening while you are dreaming the dream… Waking up and realizing that the feeling is not there is also a bit disturbing… What can you do in life is life, it’s not a dream. You have to face it…”
Guilt	P10	“Who would leave their family, kids, and everything they have? You will only leave if the threat is directed against you and your life… By leaving, you are doing the right thing at the same time opening a possibility of feeling guilty of leaving everything”.
	P2	“…You regret to some extent… But what different choices do [I] we have… It was a matter of life and death from the beginning until now. So, you will learn how to control your regrets, if you want to move forward…”
Structuring of the past in the homeland	P5	“I carefully brought almost everything I thought would be sentimental [pictures, book I wrote and other books, family home videos]. It gives me comfort when I see them… It tells me and my children that I was indeed somebody, someone important… They [sentimental things] especially help me when the work environment [working as a low skilled laborer] belittles you.”
Personal experiences of death and funerals	P4	“I lost a close family relative recently and it kills me not to be with my family in this hard time… No one told me she was sick and the news disturbed me… I don’t even know if I mourned properly.”
	P3	“ I lost my mother last year… It was the saddest time of my life and it’s even hard to know that you can’t go back and attend the funeral… I was able to mourn properly because of the Ethiopian community here. Most people came to my house to comfort me. They brought food, gave me money, and most of them stayed with me for several days. I don’t know what would happen to me if there were no one…”
Anxieties, morbid thoughts and anger in response to separation from the homeland	P2	“It was my wish to be with everyone and everything I knew but what can I do about it. I feel angry the most when I realize that I don’t belong here…”
The comfort derived from religious beliefs	P8	“It always comforts me to know that my life is in the hands of God…”
	P3	“The grace of saint Mary and Her Son is the reason why I’m standing today…”
The comfort derived from participation in religious gatherings	P4	“… You could at least chat with your fellow Ethiopians with your native language, see an Ethiopian face, and hear Ethiopian music”

## Data Availability

The data presented in this study are available on request from the corresponding author. The data are not publicly available due to ethical reasons.

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
