# Peer review of "Cultural Bereavement and Mental Distress: Examination of the Cultural Bereavement Framework through the Case of Ethiopian Refugees Living in South Korea"

_healthcare, 2022, doi:10.3390/healthcare10020201_

Round 1
Reviewer 1 Report
Thank you for this excellent revision. The manuscript is well-written, methods and results are clear and the authors have addressed my previous concerns. The interpretations from the results are sound.
One sentence struck me a bit: In the Guilt section (p9): "Though with some of the participants, there is a sense of hope that they will make things right someday". If I may offer my view that it is not 'a fault' of the refugees to leave their country, as they describe, it was for safety reasons. "Making things right" may indicate they have done something wrong. You may want to consider rewording this slightly. Possibly stating: "Though with some of the participants, there is a sense of hope that they can reconnect with people they have left behind".
Secondly, I am not sure about the following theme label: "Clarity with which appearance of relations is recalled" (p9). The question related to this theme was: "The question revolved around if they have difficulties remembering and if they do anything to stop the memories from fading." The quote provided addresses fading memories of family and homeland, I feel this fits more int the "Memories of a family in the homeland theme". Moreover, using Ethiopian TV channels and video calls was also mentioned in the "Memories of a family in the homeland theme". I strongly feel these two questions can be combined into one theme.
Thirdly, it would be helpful to read how South Korean people morn when a close relative or friend dies. It is very well described how Ethiopian people morn (p11), and it is suggested it is different from South Korean people, but not explicitly. Not being aware of their culture, it would help for readers like me to describe this shortly, as I don’t want to make any wrong assumptions.
Minor change:
- “This section of the interview intended to understand how these refugees remember their past” (p10). Remove the word "these" before refugees and change into “Ethiopian refugees” (it feels more respectful).
Reviewer 2 Report
The research question is explicit and the objectives have been clearly identified and their relevance to the clinical and preventive approach to cultural bereavement distress is well justified.
An assessment criterion for qualitative ethnographic studies is logical consistency with the methodology and results of previous similar research, a condition that this work fulfills.
The main threat to the logical consistency of the study is the biases that may be introduced by the researcher in the systematization of the recording of information during the structured interview and its analysis. In this sense, the authors have taken many measures to avoid these; for example, they were flexible with the participants so that they could give as many answers as they wished, and they have used recognizable software to analyze and code them.
The above measures affect the credibility of the results. This is achieved through prolonged observations and conversations with participants, but also by checking whether the findings obtained are recognized by the informants as representative of what they think and feel, something that could have been done. Therefore, it would have been desirable to acknowledge such a limitation in the discussion. Likewise, a deeper reflection on the relationship between the researcher and the object of the research would have been desirable. All in all, we believe that the results of the study are valid because the selection strategy of the interviewees and the data collection technique are congruent with the question and the method used. In addition, all the required ethical aspects have been taken into account.
Finally, I would like to point out that the data analysis is adequate and the results are presented in a clear way and are applicable to develop and design interventions to reduce the distress of this type of people.
Author Response
Please find the attached file

This manuscript is a resubmission of an earlier submission. The following is a list of the peer review reports and author responses from that submission.
Round 1
Reviewer 1 Report
This paper describes a qualitative study conducted in Ethiopian refugees in South Korea. I believe the topic addressed is of importance and very relevant. However, there are some concerns I have with the presentation of the study, which I have addressed per section:
Introduction:
- There are some references missing after certain statements. For example you stated: "Whereas, in the second stage (physical relocation) and the third stage of post-migration, mental stress will build up and reach a critical point." This should be cited and backed-up by evidence or be stated as an hypothesis (i.e., "... mental stress may build up"). There are some more examples throughout your introduction.
- There is no rational for why you choose Ethiopian refugees and not another ethnic group. The reasons you provide are applicable to all ethnic minorities in south Korea.
Method:
- I do like the fact that you co-designed the questions with someone from the population. However, I am concerned that the interview schedule you developed consist of leading questions. You start some questions with how many people like them feel and then ask how they feel. People do not like to be perceived as different or "weird" and could have, therefore, answered these questions in a certain way to confirm and belong to the norm.
Results:
- You start with stating that the quotes will be presented in the table and you continue presenting them in the text as well as in the table. I am not sure what the table adds to the themes and quotes presented in text.
- You are very interpretative in your result section. Either you only resent the results without any interpretation and leave this for the Discussion section or you combine you Results and Discussion section into one section.
Discussion:
- You state that the "framework is more or less able". Given that one of your objectives was to establish whether the framework is useful for the refugee population, this conclusion is not good enough. What were the benefits or challenges using this framework and what do you conclude, was it or was it not helpful and would you recommend any changes when others would like to use it in this setting?
- Please, do not use a dot-point list as your discussion; flesh out the findings, link them with previous work and interpreted the possible implications and future directions.
Reviewer 2 Report
In the introduction, the authors clearly describe the theoretical concept of cultural bereavement and state why it is worth focusing the research question on that. They also - correctly - show, that the concept of cultural bereavement is not very often used as explanatory model in refugee research. The authors also describe the context of Ethiopian refugees in South Korea. Nevertheless, it would be good to stress the explanatory power of the concept of cultural bereavement in relation to other theoretical concepts. Why does this concept better fit than others?
In general, a qualitative research design is appropriate for this research question. It would be good to better describe why the authors decided on the specific eligibility criteria. (e.g. why do participants have to have lived at least a year as a recognized refugee?) The authors do not mention any ethical considerations (e.g. how to deal with flashbacks/trauma during and/or after the interview).
The data collection and data analysis strategy could also be more elaborated on. Especially the data analysis should be more theory-based.
The results are quite clearly presented, but stays rather descriptive (what is acceptable for the quite exploratory study design of this paper). The authors draw a rather general conclusion for policy makers and practitioners. The research focus of this paper is of high practical relevance and therefore would need further elaboration.
Reviewer 3 Report
Review of paper: Understanding the mental health cost of Cultural Bereavement for refugees: Implications for policy and practice
(Healthcare - Manuscript ID: healthcare-991617)
I like starting point idea with this ms. enormously! I would like to see a sharper integrations of the key ideas. I completely missing the discussion of the importance of connection between theory, previous research, methodology and empirical data. I don't see these ideas as well integrated within the theoretical framework, previous research, methodology and empirical data, as I really want.
The article focuses on very important themes and the article itself is an important contribution to the academic studies on “migration experiences and their relevance to cultural bereavement”. My main arguments concern the theoretical framework of the article, previous research, the method, the structure and the analysis. The article needs to be revised in a serious manner. The article lacks a chapter that would critically go through previous research done in the same field and a chapter that would more broadly and deeply go through the theoretical concepts of the study.
There are a few areas in which the author could undertake some further work.
- The theoretical framework underpinning this work needs greater clarity and description. Even though it is evident that the article deals with “migration experiences and their relevance to cultural bereavement” - it is an interesting choice of the author not to use the concept of this theoretical starting points. There is a vast field of academic research concentrating “migration experiences and their relevance to cultural bereavement”. Review scientific articles on the same or similar social phenomena written as a product of research conducted in other countries. See for example scientific articles from Swedish context: Inclusion and obstacles in the Swedish social pedagogical context: an analysis of narratives on working with unaccompanied refugee minors with wartime experiences in institutional care; Collaboration and identity work: A linguistic discourse analysis of immigrant students’ presentations concerning different teachers’ roles in a school context; Ethnic monitoring and social control: Descriptions from juveniles in juvenile care institutions…
- The previous research underpinning this work is missing. Even though it is evident that the article deals with “migration experiences and their relevance to cultural bereavement”, it is an interesting choice of the author not to use and not even refer to studies done in this field. Even here, there is a vast field of academic research concentrating “migration experiences and their relevance to cultural bereavement”. The author does not argument why this kind of choice has been made. I would recommend the author to frame the article within the theoretical discussions and link this with the analytical concept (previous research) concerning “migration experiences and their relevance to cultural bereavement”. Now the hypothesis of the paper are not convincing enough and actually the author does not make much use of these concepts in the analysis itself.
- The methodological requires discussion. How were the interviewees selected? What was the criteria? What was involved in the methodological process? How were interviews processed? How were analyzed? What was the process of coding of interviews? How theoretical concepts and previous research were used in the analysis? How are interview quotes used in the analysis? There is a vast field of methodological literature that can be used in further development work with article. See for example: The SAGE handbook of qualitative data analysis; Interpreting Qualitative Data ...
- Conclusion requires a more developed synthesis and discussion to bring together the key contribution of this paper. The conclusion of the article does not bring the results and analysis on a more abstract level. How does the research contribute to the discussion of “migration experiences and their relevance to cultural bereavement”? What were the limitations of the research? What could have been done differently? How should future research tackle these questions? What about methodological questions?
- Overall, this is a good paper in progress but it could do with further clarity, focus and development.